

# Energy flow in men's javelin throw and its relationship to joint load and performance

Hans-Peter Köhler and  Maren Witt

Department of Biomechanics in Sports, Sport Science Faculty, Universität Leipzig, Leipzig, Germany

## ABSTRACT

**Background**. Performance in javelin throwing is dependent on the release speed and therefore the energy transferred to the javelin. Little is known about the flow of mechanical energy in javelin throwing and whether there is a connection to joint loading and throwing performance. The purpose of the study was therefore to investigate (1) the energy flow within the kinetic chain of the throwing arm, (2) how it is related to performance and joint loads and (3) how joint forces and torques are used to transfer, generate and absorb mechanical energy.

**Methods**. The kinematics of 10 experienced javelin throwers were recorded using a 12-camera infrared system. 16 markers were placed on the athlete's body, five on the javelin to track the movement of each segment. A segmental power analysis was carried out to calculate energy flow between upper body, upper arm, forearm and hand. Stepwise regression analysis was used to calculate the variable that best predicts release speed and joint loads.

**Results**. The results indicate that the higher the peak rate of energy transfer from the thorax to the humerus, the higher the release speed and the joint loads. While there were no differences between the peak rate of energy transfer in the different joints, the energy transferred differed depending on whether joint forces or torques were used. It can be further shown that higher joint torques and thus higher rotational kinetics at the shoulder are linked to higher release speeds. Thus, the movements of the upper body can be of great influence on the result in javelin throwing. Furthermore, the data show that athletes who are able to transfer more energy through the shoulder, rather than generate it, experience a smaller joint loading. An effective technique for improved energy transfer can thus help perform at the same level while lowering joint stress or have higher performance at the same joint loading.

## INTRODUCTION

The most important factor in javelin throwing in order to reach longer throwing distances is the release speed. While other factors like the release angle, angle of attack and angle of yaw are also important and should be optimized according to the surrounding conditions, the release speed is the only factor that has to be maximized (*Bartonietz, 2000*; *Krzyszkowski & Kipp, 2021*). Thereby, the release speed depends on the mechanical work done on the javelin by the thrower; the more work that is done, the higher the kinetic energy and

Corresponding author
Hans-Peter Köhler, hans-peter.koehler@uni-leipzig.de

thus the release speed (*Bartonietz, 2000*). The energy needed to change the javelins kinetic energy can either be generated by the working muscles or be transferred from the heavier proximal segments through the kinetic chain.

Due to the run-up in javelin throwing, a certain amount of kinetic energy is already present within the thrower at the beginning of the delivery and can therefore be transferred to the javelin. Furthermore, higher run-up velocities are associated with longer throwing distances (*Salo & Viitasalo, 1995*; *Murakami et al., 2006*). Although large amounts of energy are already present at the beginning of the delivery, in javelin throwing the final acceleration of the javelin is seen to be done partly by driving mechanisms generating mechanical energy or reusing stored elastic energy at the shoulder and elbow (*Bartonietz, 2000*; *Tidow, 2008*; *Strüder, Jonath & Scholz, 2013*). In contrast, in baseball pitching it is known that the transfer of mechanical energy from the heavier proximal segments is the most important source of energy for ball acceleration, whereas the mechanical energy generated at the throwing shoulder, elbow and wrist has little to no effect (*Howenstein, Kipp & Sabick, 2019*; *Wasserberger et al., 2021*). Therefore, the question arises regarding the extent to which is necessary to generate energy at the joint by the working muscles of the throwing arm when large amounts are already present in javelin throwing.

It is known that 70–75% of the work on the javelin has to be done in the last 100–120 ms and due to this limitation of time the mechanical power becomes the most important factor for increasing performance (*Bartonietz, 2006*; *Lehmann, 2010*). This also puts high demands on the production of high forces and torques in order to generate or transfer large amounts of mechanical energy and to secure the joint structures at the same time. Eccentric muscle contractions prior to concentric contraction are seen as essential to increase the torques needed to accelerate the implement as the time is not long enough to produce the high concentric torques and forces needed (*Jöris et al., 1985*). Thereby, high eccentrical loads are associated with an increased risk of injury (*Fleisig et al., 1995*; *Anz et al., 2010*). The loads which have gained the most attention in baseball pitching are the shoulder internal rotation and the elbow varus torque. Furthermore, structures associated with these torques in javelin throwing are frequently injured (*Schmitt et al., 2001*). Nevertheless, the joint loads acting on the joints of the throwing arm have never been reported.

While most studies rely on kinematical analysis, advanced techniques like energy flow (EF) have never been used in javelin throwing despite the common use of terms like energy transfer and generation (*Bartonietz, 2000*; *Campos, Brizuela & Ramón, 2004*; *Tidow, 2008*). However, the number of investigations using EF analysis has significant increased recently, especially in baseball pitching. EF analysis is a method that can be used to study the flow of mechanical energy between segments connected by a joint. On the one hand, the transfer of mechanical energy can be calculated, but additionally it is possible to quantify the absorbed or generated mechanical energy. It is also possible to determine the contributions of rotational and linear kinetics which arise from net joint moments and forces, respectively (*Howenstein, Kipp & Sabick, 2019*). Recently, EF analysis was used to investigate predictors for pitching performance and joint loading of the upper extremities in baseball pitching (*Aguinaldo & Escamilla, 2019*; *Wasserberger et al., 2021*) as well as mechanical patterns of the lower extremities (*Howenstein, Kipp & Sabick, 2020*;
*Pryhoda & Sabick, 2022*). The different investigations have shown that EF analysis is a highly suitable method for quantifying energetical patterns between adjacent segments and therefore provide deeper insights into the mechanics of the kinetic chain and the underlying movement (*Martin et al., 2014*).

The aim of the present study was to investigate the driving mechanisms of the different joints of the throwing arm and, thus, to gain deeper insights into the mechanics of javelin throwing using EF analysis. The study examined how the different joints utilize joint torques and forces for energy transfer, generation and absorption. Furthermore, the relation between EF variables and (A) the release speed of the javelin and (B) to the joint load were investigated.

## MATERIAL AND METHODS

### Participants
Ten right-handed javelin throwers (body height: $189.2 \pm 7.2$ cm; body mass: $92.4 \pm 9.3$ kg; age: $21.8 \pm 3.6$ years; personal best: $78.23 \pm 11.38$ m) of the German (junior-) national team participated in the study. All subjects were free from injuries at the time of the investigation. The study was approved by the Leipzig University ethics committee (ethical approval nr: 462/18-EK) and performed according to the declaration of Helsinki. All participants gave written informed consent to participate in the study prior to the investigations.

### Material and experimental protocol
The three-dimensional position data of the markers attached to the skin of the athletes were recorded at 300 Hz using 12 infrared cameras (Qualisys AB, Gothenburg, Sweden). In addition, two perpendicular video cameras (Qualisys AB, Gothenburg, Sweden) recorded the throws at 150 Hz. The camera system was set up in an oval around the last 10 m before and 2 m behind the foul line, with half of the infrared cameras on each side of the approach. The video cameras were positioned in such a way that one camera was orthogonal to the approach at the release point about two meters before the foul line. The second video camera stood 10 m away from the foul line, recording the release from the back. The average residual of the calibration was 0.75 millimeter.

To record the movements of the thrower's torso and upper extremities, 16 markers (metacarpophalangeal joint of the 2nd and 5th finger; ulnar and radial styloid; lateral and medial epicondyle of the humerus; left and right acromion; 7th cervical vertebrae and 12th thoracic vertebrae; processus xiphoideus; incisura jugularis; left and right spina iliaca anterior superior; left and right spina iliaca posterior superior) and two clusters (upper arm, forearm) were placed on each subject. Five markers were attached to the javelin (GETRA Kinetic, 800 g, 70 m), which was modified for indoor use by replacing the sharp metal with a dull carbon tip. The investigations were carried out indoors, with the athletes throwing into a net.

After an individual warm-up, each participant threw at least three trials using their preferred approach (mean approach speed: $5.05 \pm 0.62$ ms$^{-1}$). The best three throws were selected for further analysis based on the highest release speed of the javelin ($v_0$).

## Data processing

Prior to further data analysis three critical events were identified visually from the recorded videos: (1) touchdown of the rear leg, (2) touchdown of the bracing leg and (3) release of the javelin. Afterwards, marker trajectories were filtered using a fourth order, zero-lag Butterworth filter. The cut-off frequencies (10–13 Hz) for each marker were determined *via* residual analysis (*Winter, 2009*).

To calculate the kinematics and kinetics, a six-segment model (javelin, right hand, forearm, upper arm, thorax, abdomen) was built using Visual 3D (Ver. 2020.11.2; C-motion, Germantown, USA). The shoulder joint center was determined using functional methods implemented into Visual 3D (*Schwartz & Rozumalski, 2005*; *Howenstein, Kipp & Sabick, 2019*). The joint centers of the elbow and the wrist were respectively determined as the midpoints between the medial and lateral humeral epicondyles and the midpoints between the ulnar and radial styloid (*Wu et al., 2005*). In dynamic movements the pose of each segment was estimated using the six degrees of freedom algorithm. The kinetics were calculated using Newton-Euler equations of motion implemented in Visual 3D. All derived net joint forces (NJF) and torques (NJT) were calculated as internal torques and forces. For the inverse dynamics calculations, body segments inertial parameters reported by *De Leva (1996)* were used. The BSIP of the javelin were estimated using a torsion pendulum and a reaction board (*Sommerfeld, 1950*).

Using the calculated kinematics and kinetics, a segment power (SP) analysis was carried out for all segments of the different joints(shoulder, elbow, wrist) of the throwing arm using MATLAB (Ver. 9.11.0; The Mathworks Inc., Natick, MA, USA). The joint force power (JFP) and the segment torque power (STP) for the proximal and the distal segment of a joint were calculated as:

$$\text{JFP} = \boldsymbol{F_{ij}} \cdot \boldsymbol{v_j},$$
$$STP = \boldsymbol{T_{ij}} \cdot \dot{\boldsymbol{\theta}}_{ij}$$

where $\mathbf{F_{ij}}$ is the NJF vector and $\mathbf{T_{ij}}$ the NJT vector acting on the i[th] segment of the j[th] joint. $\mathbf{v_j}$ is the linear velocity vector of the j[th] joint, $\dot{\boldsymbol{\theta}}$ represents the angular velocity vector of the i[th] segment at the j[th] joint. The NJF as well as the NJT vector of both segments connected by a joint are equal in magnitude and opposite in direction. While the linear velocity is the same at the joint for both segments and therefore the rate of energy loss of one segment equals the rate of energy gain of the second segment connected with the joint, the angular velocity of the two connected segments is not necessarily the same. Hence, the power produced by the NJT can not only transfer energy, mechanical energy can also be absorbed or generated by the muscles (*Robertson & Winter, 1980*) (Table 1). To further estimate power transfer, generation and absorption (TGA), the previously calculated terms of the SP analysis were used. To account for the net rate of energy transfer (PT) at the different segments, the sum of the JFP of the distal segment of the joint and the portion of the distal STP which represents transfer by the NJT according to the conditions outlined in Table 1 was calculated (*Wasserberger et al., 2021*). Additionally, the time series for the

**Table 1  Calculation of the transfer, generation and absorption of mechanical energy.** Breakdown of the segment torquer power (STP) of two adjacent segments of a joint, based on magnitude and sign, into transfer, generation and absorption of mechanical energy via net joint torques. Adapted from *Wasserberger et al. (2021)*.

| | | Generation | Absorption | Transfer |
|---|---|---|---|---|
| **Same sign** | | | | |
| Both positive | | To proximal Segment at $T_p\dot\theta_p$<br>To distal Segment at $T_d\dot\theta_d$ | 0 | 0 |
| Both negative | | 0 | From proximal Segment at $T_p\dot\theta_p$<br>From distal Segment at $T_d\dot\theta_d$ | 0 |
| **Opposit Sign** | | | | |
| $\|STP_p\| >$ | $\|STP_d\|$ | | | |
| + | − | To proximal Segment at $T_p(\dot\theta_p - \dot\theta_d)$ | | To proximal Segment at $T_d\dot\theta_d$ |
| − | + | 0 | From proximal Segment at $T_p(\dot\theta_p - \dot\theta_d)$ | To distal Segment at $T_d\dot\theta_d$ |
| $\|STP_p\| <$ | $\|STP_d\|$ | | | |
| + | − | | From distal Segment at $T_d(\dot\theta_d - \dot\theta_p)$ | To proximal Segment at $T_p\dot\theta_p$ |
| − | + | To distal Segment at $T_d(\dot\theta_d - \dot\theta_p)$ | 0 | To distal Segment at $T_p\dot\theta_p$ |

**Notes.**
STP$_p$, segment torque power of the proximal segment; STP$_d$, segment torque power of the distal segment; $T_p$, proximal joint torque; $T_d$, distal joint torque; $\dot\theta_p$, angular velocity of the proximal segment; $\dot\theta_d$, angular velocity of the distal segment.

rate, at which energy is generated (PG) and absorbed (PA) at the corresponding joint was calculated accordingly.

Beside the EF between the different segments, the kinematics and the kinetics of the center of mass (CoM) of the javelin were also calculated. The position of the CoM of the javelin was calculated based on data obtained from markers attached to the javelin. The velocity ($v_j$) of the javelin's CoM was calculated *via* differentiation of the CoM's position. The release speed ($v_0$) of the javelin was defined as the magnitude of the javelin's velocity one frame after release. Furthermore, the acceleration force ($\mathbf{F}_{acc}$) and acceleration power ($P_{acc}$) acting on the CoM of the javelin were calculated. The $\mathbf{F}_{acc}$ was calculated as:

$$\mathbf{F_{acc}} = \mathbf{a_j} m_j = \mathbf{\dot{v}_j} m_j,$$

where $\mathbf{a}_j$ is the acceleration vector of the javelin's CoM and $m_j$ is the mass of the javelin. The $P_{acc}$ of the javelin was calculated as:

$$P_{acc} = \mathbf{F_{acc}} \mathbf{v_j}.$$

Using the SP power-time series, peak segment torque power (pPST) and peak joint force power (pJFP) acting on the proximal end of the distal segment of a joint were extracted. From the TGA power-time series, peak transfer (pPT), generation (pPG) and absorption (pPA) were extracted for each joint. In addition, by integrating the different power-time series from the touchdown of the rear leg until the release of the javelin, the energy that was transferred, generated or absorbed by the TGA power terms and the energy delivered by the NJT and NJF to the distal segment was calculated. The peak shoulder internal rotation ($T^{IR}$) and elbow varus torque ($T^{VAR}$) were extracted from the joint moments. Furthermore, the peak values were extracted for both $F_{acc}$ and $P_{acc}$.

## Statistics

All extracted variables were tested for normal distribution using the Shapiro–Wilk test. A repeated measures analysis of variance (rmANOVA) was conducted for the different peak power and energy terms of the shoulder, elbow and wrist joint to investigate joint differences. In addition, Mauchly's test of sphericity was carried out to test for equal variances of differences. If the assumption was violated, Greenhouse-Geisser correction was used to adjust the degrees of freedom. If a main model effect ($p < 0.05$) was found, Bonferroni-corrected post-hoc comparisons were calculated.

To test for relationships between the power variables and $v_0$, $T^{IR}$ and $T^{VAR}$, a correlation analysis was calculated. Afterwards, the extracted peak Power variables, separated by TGA and SP, were entered as independent variables into a multiple stepwise regression analysis to determine the linear model that best predicts $v_0$, $T^{IR}$ and $T^{VAR}$, respectively. Power variables were normalized using body mass (BM), the $T^{IR}$ and $T^{VAR}$ were normalized by BM × body height (BH). As for the rmANOVA, the normalized data were tested for normal distribution. To address multicollinearity between predictor variables, the variance inflation factor (VIF) was observed. In case of collinearity (VIF > 10), the most distal terms were removed first in order to receive the earliest predictor possible. The model was chosen by an information theory approach using corrected Akaikes Information Criterion (AICc), which was corrected for small sample sizes (*DelSole & Tippett, 2021*). The model with the lowest AICc was chosen for TGA and SP inputs. To compare models between TGA and SP, the goodness of fit for each model was described by the adjusted coefficient of determination ($R^2$). Unstandardized regression model parameters (B) as well as their 95% confidence intervals (95% CI) were calculated. All statistical analyses were conducted using MATLAB.

## RESULTS

The athletes reached a mean $v_0$ of $22.73 \pm 1.28$ ms$^{-1}$: The mean peak $F_{acc}$ reached $213 \pm 30$ N and the mean peak $P_{acc}$ reached $2939 \pm 506$ W (Fig. 1A). Thereby a mean $T^{IR}$ of $117 \pm 27$ Nm ($67.19 \pm 13.68\%$ BM×BH) and a mean $T^{VAR}$ of $116 \pm 26$ Nm ($66.68 \pm 13.81\%$ BM×BH) could be calculated *via* inverse dynamics.

The different power–time series can be seen in Figs. 1B–1E. For the pJFP ($F_{2,18} = 57.26$; $p < .001$; $\eta_p^2 = 0.864$) as well as the pSTP ($F_{2,18} = 85.54$; $p < .001$; $\eta_p^2 = 0.905$) the rmANOVA revealed a significant main model effect between the different joints (Fig. 2A). Joint differences were also found between the subdivided pPG ($F_{2,18} = 14.17$; $p < .001$; $\eta_p^2 = 0.612$) and pPA ($F_{2,18} = 7.15$; $p = .005$; $\eta_p^2 = 0.443$), whereas the pPT ($F_{2,18} = 1.37$; $p = .280$; $\eta_p^2 = 0.132$) showed no significant differences between the joints (Fig. 2B).

In addition, for energy entering the distal segment using the STP ($F_{2,18} = 328.71$; $p < .001$; $\eta_p^2 = 0.973$) and JFP ($F_{2,18} = 130.90$; $p < .001$ $\eta_p^2 = 0.936$) in the different joints, significant main model effects were found (Fig. 2C). In case of the divided power terms, the energy which was transferred ($F_{2,18} = 88.32$; $p < .001$; $\eta_p^2 = 0.908$), generated ($F_{1.22,11.03} = 28.97$; $p < .001$; $\eta_p^2 = 0.763$) and absorbed ($F_{2,18} = 34.78$; $p < .001$ $\eta_p^2 = 0.794$) at the various joints, differed significantly (Fig. 2D).

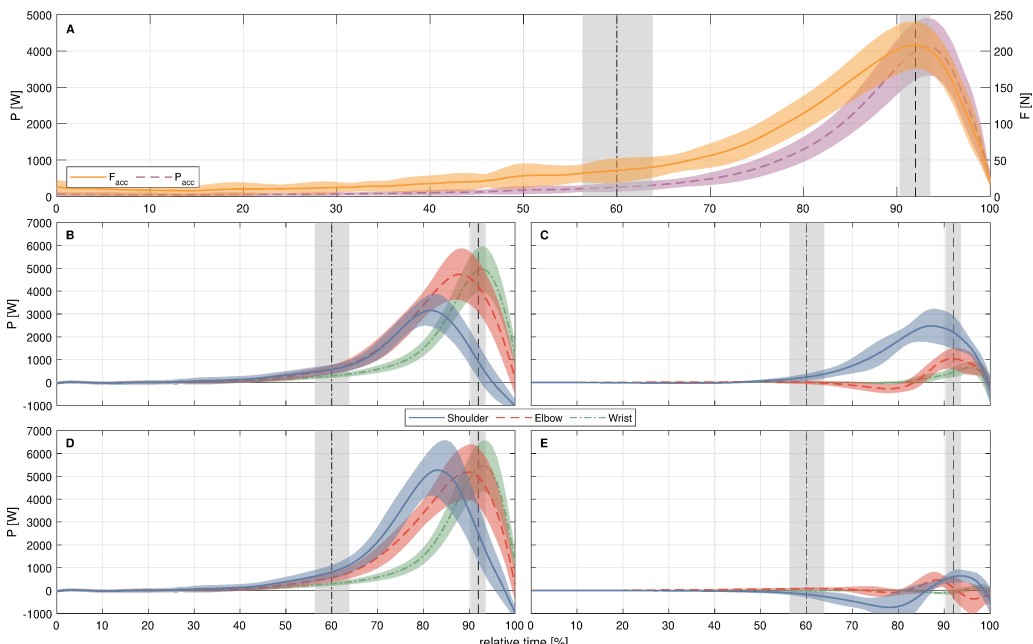

**Figure 1  Normalized time histories for the various power variables acting at the different joints and canter of mass of the javelin.** (A) Acceleration force ($F_{acc}$) and acceleration power ($P_{acc}$) acting on the center of mass of the javelin, (B) joint force power ($P_{JF}$) acting at the different joints, (C) segment torque Power ($P_{ST}$) acting at the distal segment of the considered joint, (D) rate at which energy is transferred at the different joints, for (B–D) positive values indicate energy flow to the distal segment. (E) shows the time series of energy absorption (negative) and generation (positive) at the different joints. All values are shown as $jx \pm s$, 0% represents the touchdown of the rear leg, 100% the release of the javelin, the dash-dotted line and grey area represent the $jx \pm s$ of the touchdown of the bracing leg, the dashed line with the grey area represents the $jx \pm s$ of the maximum external rotation of the shoulder.

$P_{acc}$ ($r = .958$; $p < .001$; 95% CI = [.825;.990]) and $F_{acc}$ ($r = .799$; $p = .005$; 95% CI = [.340;.950]) were strongly correlated to $v_0$. Furthermore, several of the different peak power terms were correlated to $v_0$, $T^{IR}$ and $T^{VAR}$ respectively (Table 2). The stepwise linear regression was able to predict $v_0$, $T^{IR}$ and $T^{VAR}$ using the TGA as well as the SP analysis (Table 3). While the linear regression using the TGA values could predict 79,1%, 50,5% and 49,5% of the variance in $v_0$, $T^{IR}$ and $T^{VAR}$, respectively, the SP model showed better predictions ($v_0$:87,6%, $T^{IR}$: 72,9%; $T^{VAR}$: 69,1%) while reaching lower AICc values.

## DISCUSSION

This study is the first to investigate the flow of mechanical energy within the throwing arm in javelin throwing. The aim was to investigate the driving mechanisms at the different joints used in men's javelin and its relation to performance and joint load.

### Energy flow and performance

The results indicate that mechanical energy is primarily transferred across the joints of the upper arm, which is in concordance with findings from baseball studies (*Aguinaldo & Escamilla, 2019*; *Howenstein, Kipp & Sabick, 2019*; *Wasserberger et al., 2020*). While the

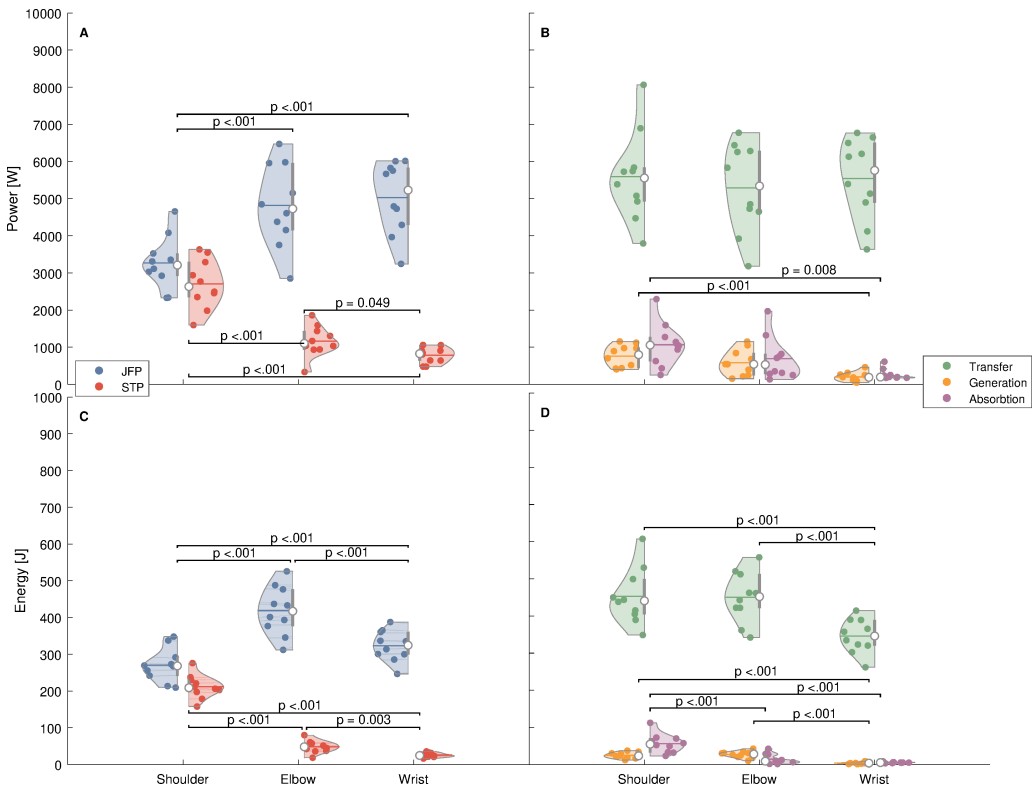

**Figure 2** **Violins plots for the differences between the different joints regarding peak power and energy delivered to the distal segment.** Peak power (A) and energy (C) delivered to the distal segment by the joint forces (JFP) and segment torques (STP) for the shoulder, elbow and wrist joint. Peak rates of energy transfer, generation and absorption (B) and energy transferred, generated and absorbed (D) by the different joints. All values are shown as $\bar{x} \pm s$. Additionally, significant differences were marked with the corresponding $p$-values for the different power and energy terms between the different joints.

production of small amounts of energy at the different joints cannot not be denied, it cannot be stated conclusively whether this energy is reused elastic energy or energy generated from the muscles. However, while the correlation analysis could not reveal relations between the $v_0$ and the peak energy generation at the different joints, the best TGA regression model, chosen by the AICc, indicates that the peak power generation at the shoulder is related to $v_0$. This observation is redundant with the SP regression model, where peak joint force power and peak segment torque power at the shoulder entered the regression, the latter containing transfer and generation of energy by the NJT. Therefore, the results indicate that the shoulder is the last joint inserting energy into the kinetic chain, which partly confirms the previous assumptions made in javelin throwing, but neglects the proposed need for energy generation at the elbow and wrist joint (*Bartonietz, 2000*; *Tidow, 2008*; *Strüder, Jonath & Scholz, 2013*).

The need for high rates of energy transfer is also supported by the different time series, as well as the differences in the energy transferred by the different power terms. The $P_{acc}$ peaks shortly after reaching the timepoint of the maximum external rotation(MER) at the

**Table 2  Results of the Pearson-correlation analysis.** Pearson correlation coefficients ($r$), level of significance ($p$) and the 95% confidence interval of the correlation coefficient (95% CI) for the correlation of the different normalized peak power terms at the shoulder, elbow and wrist joint with the release speed, normalized shoulder internal rotation and normalized elbow varus torque. Significant correlations are marked in italic.

| | | Release speed | | | Normalized shoulder internal rotation torque | | | Normalized elbow varus torque | | |
|---|---|---|---|---|---|---|---|---|---|---|
| | | r | p | 95% CI | r | p | 95% CI | r | p | 95% CI |
| Shoulder | pJFP | .719 | .019 | [.165 ;.929] | .468 | .173 | [−.230;.848] | .465 | 0.176 | [−.233;0.847] |
| | pSTP | .863 | .001 | [.513 ;.967] | .871 | .001 | [.535 ;.969] | .852 | 0.002 | [.479 ;0.964] |
| | pPT | .837 | .003 | [.439 ;.961] | .614 | .059 | [−.026;.897] | .603 | 0.065 | [−.043;0.893] |
| | pPG | .233 | .516 | [−.464;.752] | .503 | .138 | [−.185; .860] | .509 | 0.133 | [−.178;0.862] |
| | pPA | .317 | .372 | [−.390;.789] | −.536 | .110 | [-.872; .141] | .566 | 0.088 | [−.099;0.881] |
| Elbow | pJFP | .825 | .003 | [.406 ;.957] | .807 | .005 | [.361 ;.953] | .789 | 0.007 | [.317 ;0.948] |
| | pSTP | .447 | .196 | [−.255;.840] | .559 | .093 | [−.109;.879] | .539 | 0.108 | [−.137;0.873] |
| | pPT | .841 | .002 | [.451 ;.962] | .876 | .001 | [.548 ;.970] | .857 | 0.002 | [.495 ;0.966] |
| | pPG | .366 | .298 | [−.342;.809] | .424 | .222 | [−.281;.832] | .474 | 0.166 | [−.222;0.850] |
| | pPA | −.243 | .498 | [−.757;.456] | .200 | .579 | [−.491;.737] | .238 | 0.509 | [−.461;0.754] |
| Wrist | pJFP | .839 | .002 | [.444;.961] | .852 | .002 | [.480 ;.964] | .852 | 0.002 | [.479 ;0.964] |
| | pSTP | .705 | .023 | [.136;.924] | .694 | .026 | [.114 ;.921] | .646 | 0.044 | [.028 ;0.907] |
| | pPT | .819 | .004 | [.391;.956] | .761 | .011 | [.253 ;.940] | .752 | 0.012 | [.232 ;0.938] |
| | pPG | .562 | .091 | [−.105;.880] | .628 | .052 | [−.003;.901] | .598 | 0.068 | [−.050;0.892] |
| | pPA | .317 | .372 | [−.390;.789] | −.358 | .310 | [−.806;.351] | −.352 | 0.319 | [−.803;0.357] |

**Notes.**
pJFP, peak joint force power; pSTP, peak segment torque power; pPT, peak rate of energy transfer; pPG, peak rate of energy generation; pPA, peak rate of energy absorption.

shoulder, therefore the energy flow to the javelin up to MER seems to be realized by the transfer from the proximal segments. As a shortening of the phase up to MER, where the muscles contract eccentrically, would lead to a reduced release speed, the amplitude needs to be maintained (*Roach & Lieberman, 2014*). To enhance the rate of energy transfer in this phase the eccentric angle-torque relationship should be improved as this would lead to higher joint torques which are basic requirements to do more work. Also, due to eccentric training not only the torque–angle relationship could be improved, fascicle lengthening could also be reached. This could be advantageous as longer fascicles are generally linked to higher power output (*Kumagai et al., 2000*; *Blazevich et al., 2007*; *Butterfield, 2010*; *Lee et al., 2021*).

If one looks at the energy generated and transferred at the shoulder, one can see that the transfer of mechanical energy accounts for a significantly larger fraction. The question is, therefore, whether and to what extent the generation of mechanical energy at the shoulder contributes to the release speed and whether a focus on the transfer of mechanical energy would be advantageous. However, this cannot be conclusively clarified with the current data.

Interestingly, the different joints do not differ in terms of peak energy transfer, but they do if STP or JFP and thus NJT or NJF are used. While both the STP and the JFP are utilized to transfer mechanical energy at the shoulder joint, the more distal joints rely on the energy transfer by the JFP. As the TGA regression models show, it is most important

**Table 3  Result of the multiple stepwise linear regression.** Unstandardized regression coefficient (B) and their 95% confidence intervals (95% CI) of the best fit forward regression models for the release speed, normalized shoulder internal rotation and normalized elbow varus torque. For each of the three dependent variables, the best fit model was calculated using the different normalized power variables separately for the segmental power analysis (SP) and the transfer, absorption, generation analysis (TGA). Of the two types of analysis, the power variables of each joint (shoulder, elbow, wrist) were used as the starting point for the forward regression.

| Intercept | | Release speed | | | | | Normalized shoulder internal rotation torque | | | | | Normalized elbow varus torque | | | | |
|---|---|---|---|---|---|---|---|---|---|---|---|---|---|---|---|---|
| | | TGA | | | SP | | | TGA | | | SP | | | TGA | | | SP | | |
| | | B | 95% CI | | B | 95% CI | | B | 95% CI | | B | 95% CI | | B | 95% CI | | B | 95% CI | |
| | | 14.96 | 11.17 | 18.76 | 14.93 | 12.33 | 17.53 | −0.070 | −0.641 | 0.501 | 0.078 | −0.199 | 0.356 | −0.074 | −0.656 | 0.508 | 0.081 | −0.219 | 0.380 |
| | pSTP | | | | 0.155 | 0.083 | 0.227 | | | | 0.020 | 0.011 | 0.030 | | | | 0.020 | 0.010 | 0.030 |
| | pJFP | | | | 0.094 | 0.008 | 0.179 | | | | | | | | | | | | |
| Shoulder | pPT | 0.128 | 0.071 | 0.185 | | | | 0.010 | 0.001 | 0.019 | | | | 0.010 | 0.000 | 0.019 | | | |
| | pPG | 0.075 | −0.056 | 0.205 | | | | 0.018 | −0.002 | 0.039 | | | | 0.019 | −0.002 | 0.040 | | | |
| Wrist | pPA | −0.189 | −0.482 | 0.104 | | | | | | | | | | | | | | | |
| Model Statistics | | adj. $R^2$ = .791; $F_{3,6}$ = 12.38; $p$ = .005; AICc = 35.47 | | | adj. $R^2$ = .876; $F_{2,7}$ = 32.72; $p < .001$; AICc = 18.94 | | | adj. $R^2$ = .505; $F_{2,7}$ = 5.59; $p$ = .036; AICc = −12.00 | | | adj. $R^2$ = .729; $F_{1,8}$ = 25.21; $p$ = .001; AICc = −20.98 | | | adj. $R^2$ = .495; $F_{2,7}$ = 5.41; $p$ = .038; AICc = −11.61 | | | adj. $R^2$ = .691; $F_{1,8}$ = 21.13; $p$ = .002; AICc = −19.47 | | |

**Notes.**

pSTP, peak segment torque power; pJFP, peak joint force power; pPT, peak rate of energy transfer; pPG, peak rate of energy generation; pPA, peak rate of energy absorption; adj. $R^2$, adjusted coefficient of determination; F, F value with degrees of freedom as indices; $p$, level of significance; AICc, corrected Akaikes Information Criterion.

to raise the rate of energy transfer into the distal segment of the shoulder: for each W/kg increased peak transfer, the $v_0$ increased by 0.13 ms$^{-1}$. But as the SP model indicates, this should mainly be done by raising the peak segment torque power, or more specifically the transfer components thereof. Each unit raise leads to an improvement in the $v_0$ of about 0.16 ms$^{-1}$, while the unit raise of the peak joint force power leads to an improvement of about 0.09 ms$^{-1}$. Therefore, the rotations of the upper body seem to be an important factor in transferring energy through the shoulder utilizing NJT, which was also found in baseball (*Aguinaldo & Escamilla, 2019*; *Aguinaldo & Escamilla, 2022*). For javelin throwing it could therefore be assumed that the rotation of the upper body about its longitudinal axis and its forward tilt are essential motions needed to transfer mechanical energy. It is therefore very important to generate a stable support, especially by means of the bracing leg, to allow the upper body to act in the desired manner (*Bartonietz, 2000*; *Tidow, 2008*).

While the peak energy transfer is the same at all joints, only the energy transferred by the wrist decreases. This could be due to the close temporal proximity of the peak energy transfer to the release and therefore the short period of time available for energy transfer. To allow for more time to transfer energy, the path travelled by the javelin within the acceleration phase should be as long as possible. As *Bartonietz (2000)* stated, the active work of the upper body towards the throwing direction that is often observed in elite throwers may help to preserve a long acceleration path and therefore allow for sufficient time to transfer mechanical energy into the javelin.

## Energy flow and joint load

The calculated joint loads for the internal shoulder rotation and elbow varus torque show values which are higher compared to recently reported values from professional baseball pitching ($T^{IR}$: 100 ± 16 Nm [59.8 ± 8.8% BM×BH]; $T^{VAR}$: 99 ± 17 Nm [59.8 ± 7.8% BM×BH]) although the release speed is comparatively lower (22.73 ± 1.28 ms$^{-1}$ *vs.* 38.1 ± 1.6 ms$^{-1}$) (*Oi et al., 2019*). Even if a comparison should be considered carefully between different research works due to the body-segmental inertia parameters and the modelling framework that were used (*Gasparutto et al., 2021*; *Sterner et al., 2022*; *Köhler et al., 2023*), it seems that the heavier javelin puts higher loads on the throwers' joints. It should also be taken into account that in competition significantly higher release speeds are achieved and with an increase in release speed, an increase in the joint load is to be expected (*Slowik et al., 2019*; *Köhler, Hepp & Witt, 2022*). This is supported by the correlation between segment torque power at the shoulder and $T^{IR}$ and $T^{VAR}$ as well as $v_0$. Furthermore, the SP regression models for the three dependent variables contain the same predictor (peak segment torque power). Thus, athletes with greater peak segment torque power have a faster release speed but are also exposed to bigger $T^{IR}$ and $T^{VAR}$, and hence, a higher load on the shoulder and elbow. Therefore, an appropriate preparation for high loads is urgently needed in order to prevent injuries and to avoid long-term absences from training and competition. Particular attention should also be paid to the specifics of the joint loads and the type of muscle contraction, since general intervention strategies seem to be ineffective in avoiding overuse injuries (*Achenbach et al., 2022*).

When using the TGA regression model, the increase in joint loads also seems unavoidable. But as the differentiation in peak energy transfer and peak energy generation at the shoulder shows, there is the opportunity to make the increase of the joint load more efficient. While the effect of peak energy generation is minor compared to peak energy transfer relating to the raise in $v_0$, this relationship is reversed for both joint loads. For each W/kg raise in peak energy generation, there is an increase of almost double for both joint loads compared to peak energy transfer. This leads to the assumption that the same $v_0$ can be reached with lower joint loads, or higher release speeds at the same load, if the thrower is able to transfer more energy from the proximal segments rather than generating it at the shoulder joint. This is in line with the findings of *Howenstein, Kipp & Sabick (2019)* for baseball pitching and *Martin et al. (2014)* for tennis. Both showed that players who took advantage of energy transfer from the proximal segments had lower joint loads or used their joint loads more efficiently. Furthermore, other studies reported that improper timings within the kinetic chain lead to reduced $v_0$ and increased joint loads (*Aguinaldo, Buttermore & Chambers, 2007*; *Urbin et al., 2013*). It was proposed by *Urbin et al. (2013)*, that the disruption in energy transfer caused by improper timing of pelvis and trunk rotation either caused a reduction in release velocity, or resulted in increased $T^{IR}$ and $T^{VAR}$ in an attempt to compensate for the lost energy. Therefore, our data also indicate that the throwing mechanics should be focused on an effective use of the joint loads by focusing on the transfer of mechanical energy within the kinetic chain rather than the generation in the upper extremity joints. This would increase performance while reducing joint loads and, therefore, reduce injury risk as well.

## Limitations

The results of the present study should be considered in the context of the following limitations. Firstly, we must take into consideration the sample size. Several statistically significant and relevant (practical and clinical) results could be found. Although the prerequisites for the selected statistical tests were checked and fulfilled, the results must be viewed with a certain degree of caution due to the group size. Nevertheless, the results show important new findings, which are also in line with the results of other sports. Therefore, they can be considered quite plausible.

Secondly, the release velocities were relatively low compared to results from competitions. This could be due to different reasons. Firstly, we should mention the timepoint of the investigation, which was several months before the competition season. Secondly, the investigation was carried out indoors, which does not meet the current conditions in javelin throwing. Furthermore, the indoor experiment leads to the fact that not the throwing distance but the release speed was used as a performance parameter. Since the competition performance depends to a large extent, but not only, on the release speed, a transferability of the results may not be possible without restrictions.

Finally, motion capturing and multi body modelling has several limitations. Errors may arise from marker motion, estimation of the body segment inertia parameters as well as the computation of joint centers. However, in the context of the chosen methods, everything possible was done to minimize their influence.

## CONCLUSION

Our study is the first to gain deeper insight into the mechanics of javelin throwing using EF analysis. It shows that the shoulder is a key point in the javelin throw. The higher the energy that is transferred through the shoulder into the throwing arm, the higher the release speed.

The effective transfer of energy also makes sure that joint loads are used efficiently. By ensuring high rates of energy transfer across the shoulder, an equivalent release speed can be achieved with lower joint loading acting on the shoulder and elbow joint. Furthermore, it could be shown that the generation of mechanical energy distal to the shoulder joint does not contribute to the generation of the release speed of the javelin. The results thus contradict some of the assumed driving mechanisms in javelin throwing. Athletes should therefore be trained technically to use the joint torques efficiently, using the kinetic chain to transfer as much energy as possible and to limit the generation of mechanical energy to reduce joint loading.

Furthermore, we were able to show that EF analysis is a very valuable tool to investigate the energy distribution within the kinematic chain in javelin throwing and to gain a better understanding of the underlying mechanics. However, the statements made depend on the chosen variant (TGA *vs.* SP) of the analysis, each of which has its own advantages.

## ACKNOWLEDGEMENTS

We would like to thank Eva Böker, Simon Kiem and Florian Hallmann for their help with data recording. We would also like to thank Boris Obergföll, Matthias Rau and Stefan Erlewein, as well as all athletes and coaches who made this investigation possible.

### Funding

This work was funded by the Federal Institute of Sports Science on behalf of the German Bundestag (No. ZMVI4-070801/19-20). This publication was funded by the Open Access Publishing Fund of Leipzig University supported by the German Research Foundation within the program Open Access Publication Funding. The funders had no role in study design, data collection and analysis, decision to publish, or preparation of the manuscript.

### Grant Disclosures

The following grant information was disclosed by the authors:
Federal Institute of Sports Science on behalf of the German Bundestag: ZMVI4-070801/19-20.
German Research Foundation within the program Open Access Publication Funding.

### Competing Interests

The authors declare there are no competing interests.

## Author Contributions

- Hans-Peter Köhler conceived and designed the experiments, performed the experiments, analyzed the data, prepared figures and/or tables, authored or reviewed drafts of the article, and approved the final draft.
- Maren Witt conceived and designed the experiments, authored or reviewed drafts of the article, and approved the final draft.

## Human Ethics

The following information was supplied relating to ethical approvals (*i.e.*, approving body and any reference numbers):

The Leipzig University ethics cimitee granted Ethical approval to carry out the study (Ethical Application Ref: 462/18-EK).

## Data Availability

The raw data is available in the Supplemental File.

## Supplemental Information

Supplemental information for this article can be found online at http://dx.doi.org/10.7717/peerj.16081#supplemental-information.

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
