# Peer review of "Energy flow in men’s javelin throw and its relationship to joint load and performance"

_PeerJ, doi:10.7717/peerj.16081_

## Round 0.1 · original submission · Minor Revisions

Further to the reviewer comments please specifically focus on the consistent use of abbreviations and as identified by two of the reviewers please ensure your interpretations do not extend beyond the scope of your results.

Reviewer 1 ·

Basic reporting

No comment

Experimental design

No comment

Validity of the findings

I only have a minor comment in relation to the interpretation of the findings. Specifically, in several instances the authors use verbs like "reduce" or "increase" in their interpretation of the statistical regression results and the practical applications of these findings e.g., in line 29 of the Abstract "The data show that an increase in the joint torques and thus an increase in the rotational kinetics of the shoulder are essential to improve performance". This interpretation, however, is not valid based on a cross-sectional study with regression methods (although I would generally agree with their intuition). I would therefore just caution the authors not to overstate their findings and would suggest that they edit the manuscript in line with this limitation (as they do nicely in the previous sentence).

Additional comments

Additional minor comments are:
1. The tables should clarify that T_IR refers to the shoulder and that T_VAR refer to the elbow. In general, the table captions/legends (e.g., Table 2) could use a bit more detail so that the reader could interpret them without having to refer back to the methods.

2. Do the authors have the measured distances of the throws? If yes, it would be nice to provide them, too, in addition to the release velocity.

3. The best 3 throws were used for analysis - were the values averaged for each thrower or did the authors include 3 trials from each thrower into the regression etc. (i.e., 30 trials total)?

4. Similar to the last comment - if trials were selected based on best "release velocity" there is a chance that these may actually not represent the *furthest* throws (i.e., best throws by competition standards). Could this affect the results and interpretation and thus be a limitation? It might be good to add this to the limitation section.

·

Basic reporting

Overall, this is a solidly written paper that adequately describes an energy flow (EF) approach in analysing the transfer, generation, and absorption between segments during javelin throwing. References are cited appropriately in the text, but I made a few recommendations of papers that would strengthen the points made by the authors in specific sections of the paper.

One general comment worth highlighting is in regard to the spurious and somewhat inconsistent use of abbreviations throughout the paper, which makes it difficult to clearly comprehend the results and the authors' interpretations of them. For example, Pst and Pjf are used to denote segment torque power and joint force power, respectively, yet they are abbreviated as STP and JFP in Figure 2. In fact, I suggest using these abbreviations (STP, JFP, JTP, etc.) instead to stay consistent with what are typically reported in the EF literature (Aguinaldo & Nicholson, 2021; Howenstein, Kipp, & Sabick, 2018; Martin et al., 2014; Wasserberger, Giordano, de Swart, Barfield, & Oliver, 2021). Furthermore, a table of all the abbreviations could help in the readability of the paper.

Lines 82-83: Technically, this particular Howenstein study did not examine the associations between EF and joint loading parameters while other papers in the current literature reported relevant associations (Aguinaldo & Escamilla, 2019; Aguinaldo & Nicholson, 2021).

Line 90: Correct typo of “torques”

Experimental design

Line 103: Add “Hz” or “fps” after “300”

Lines 103-104: Comment on the configuration and calibration residuals of the motion capture system.

Line 125: Add “respectively” before “determined”

Line 151: Add “the conditions outlined in” before “Table 1”

Line 179: List these joints (i.e., shoulder, elbow, wrist)

Line 187: Using normalized metrics as independent variables in a linear regression may violate the assumptions of normality and homoscedasticity as body mass and height could also be predictors of the response variable (Nevill & Holder, 1995). Were these assumptions confirmed?

Line 188: Specify the VIF threshold (i.e., 10) above which the assumption of multicollinearity was not tenable.

Validity of the findings

It is important to note that while the authors found linear models that best predict javelin velocity and joint torques, the mechanisms by which energy transfer, generation, and absorption at each joint cannot be defined by the analysis of this study. Regression analysis only provides the combination of energetic factors that can explain a proportion of the observed variance of a response variable (i.e., velocity), not their direct contribution to the magnitude of the variable (Aguinaldo & Escamilla, 2022). Caution should be taken not to extend the external validity of these relationships beyond the scope of their influence on performance (velo) and injury risk (joint torques).

Additional comments

References Cited

Aguinaldo, A. L., & Escamilla, R. F. (2019). Segmental power analysis of sequential body motion and elbow valgus loading during baseball pitching: Comparison between professional and high school baseball players. Orthopaedic Journal of Sports Medicine, 7(2), 232596711982792. https://doi.org/10.1177/2325967119827924

Aguinaldo, A. L., & Escamilla, R. F. (2022). Induced power analysis of sequential body motion and elbow valgus load during baseball pitching. Sports Biomechanics, 21(7), 824–836. https://doi.org/10.1080/14763141.2019.1696881

Aguinaldo, A. L., & Nicholson, K. F. (2021). Lower body contributions to pelvis energy flow and pitch velocity in collegiate baseball players. ISBS Proceedings Archive, 39(1), 137–140. Retrieved from https://commons.nmu.edu/isbs/vol39/iss1/36/

Howenstein, J., Kipp, K., & Sabick, M. (2018). Energy flow analysis to investigate youth pitching velocity and efficiency. Medicine & Science in Sports & Exercise, 51(3), 523–531. https://doi.org/10.1249/MSS.0000000000001813

Martin, C., Bideau, B., Bideau, N., Nicolas, G., Delamarche, P., & Kulpa, R. (2014). Energy flow analysis during the tennis serve: Comparison between injured and noninjured tennis players. American Journal of Sports Medicine, 42(11), 2751–2760. https://doi.org/10.1177/0363546514547173

Nevill, A. M., & Holder, R. L. (1995). Scaling, normalizing, and per ratio standards: An allometric modeling approach. Journal of Applied Physiology, 79(3), 1027–1031. https://doi.org/10.1152/jappl.1995.79.3.1027

Wasserberger, K. W., Giordano, K. A., de Swart, A., Barfield, J. W., & Oliver, G. D. (2021). Energy generation, absorption, and transfer at the shoulder and elbow in youth baseball pitchers. Sports Biomechanics, 00(00), 1–16. https://doi.org/10.1080/14763141.2021.1933158

·

Basic reporting

1. Line 68, check "internal shoulder internal rotation and ...". It should be "shoulder internal rotation".

Experimental design

1. The description in "Data processing" is not easy to follow. Please use equation or list.
2. The definitions of PT, PG, and PA should state in the main text.
3. The abbreviation in this study should be re-checked. e.g. PST: P is power; PT: P is peak rate. It is not easy to understand P is power or peak rate.

Validity of the findings

1. The authors used ANOVA test to find if the variables in joints different or not. Please state why and the application of the results.
2. Please use symbols to identify the statistical significance in table 2.
3. In table 3, it is not clear to understand whether the PST, PJF, PT, PG and PA are in shoulder or wrist joint.
4. In figure 2. what is the shape or area meaning? What is the meanings in horizontal axis of each joint?

---

## Round 0.2 · accepted · Accept

Thank you for addressing all reviewer comments comprehensively I am pleased to recommend your paper for publication.

E.g. "Line 222: It should be "VIF > 10" instead of "VIF < 10" to indicate when there could be potential cases of multicollinearity."

Reviewer 1 ·

Basic reporting

No comment

Experimental design

No comment

Validity of the findings

No comment

·

Basic reporting

No comment

Experimental design

Thank you for addressing the comments and annotated suggestions made in my initial review. The paper has improved with this revision, in my professional opinion. I only have one minor correction to be made:

Line 222: It should be "VIF > 10" instead of "VIF < 10" to indicate when there could be potential cases of multicollinearity.

Validity of the findings

Lines 403-406 now address the limitations in the external validity of the authors' results.